**communications** engineering

# Active optical phased array integrated within a micro-cantilever
Sylvain Guerber, Daivid Fowler ✉, Laurent Mollard, Christel Dieppedale, Gwenael Le Rhun, Antoine Hamelin, Jonathan Faugier-Tovar & Kim Abdoul-Carime

Three dimensional sensing is essential in order that machines may operate in and interact with complex dynamic environments. Solid-state beam scanning devices are seen as being key to achieving required system specifications in terms of sensing range, resolution, refresh rate and cost. Integrated optical phased arrays fabricated on silicon wafers are a potential solution, but demonstrated devices with system-level performance currently rely on expensive widely tunable source lasers. Here, we combine silicon nitride photonics and micro-electromechanical system technologies, demonstrating the integration of an active photonic beam-steering circuit into a piezoelectric actuated micro cantilever. An optical phased array, operating at a wavelength of 905 nm, provides output beam scanning over a range of 17° in one dimension, while the inclination of the entire circuit and consequently the angle of the output beam in a second dimension can be independently modified over a range of up to 40° using the piezoelectric actuator.

Integrated optical phased arrays (OPAs) fabricated in CMOS foundries have the potential to be used for solid-state beam steering at the heart of mechanically robust, low-cost LIDAR systems[1–5]. By individually controlling the phase of light of wavelength $\lambda$, emitted from a sufficiently high number ($10^2 < N_{ch} < 10^4$) of closely spaced ($d < \lambda$) emitters, a low divergence (<1 mRad) beam may be scanned at high frequencies (>1 MHz) over a wide (>90°) field of view (FOV) without the emergence of higher order beams and thus maintaining a high (>90%) power fraction in the main output beam[6]. Numerous proof-of-concept demonstrations based on silicon and/or silicon-nitride photonics technology have been presented over the last decade, in general using wire-bonded chips with several tens or hundreds of electrical/optical channels[7–10]. More recently, the application of more advanced electronic integration techniques shows a pathway to a low unit-cost device consisting of a one dimensional emitter array with $N_{ch} > 1000$ and $d < \lambda$, allowing beam properties compatible with demanding applications such as automotive LIDAR[11–13]. In such 1D-OPAs, by applying a set of pre-calibrated voltages to the integrated phase modulators preceding each unitary emitter, the angle of the output beam $\phi$, can be swept over an inclined plane (indicated in gray in Fig. 1a). The angle of the inclined plane to the chip surface normal, $\theta$, is a function of the parameters of the diffraction grating used to terminate each optical channel, and the operating wavelength, $\lambda$.

Beam scanning in two dimensions ($\phi$ and $\theta$) may, in principle, be achieved at a fixed wavelength using a two dimensional matrix of emitters[14–18]. However, unlike LED arrays and CMOS detector arrays that leverage dense opto-electronic integration to create large 2D arrays of emission/reception points, a phased array requires that the pitch, $d$, between

the emission apertures of adjacent optical channels be less than the operating wavelength, $\lambda$ (and as little as $d = \lambda/2$ to ensure a high power fraction emitted in the fundamental emission lobe for any main beam angle, $0 < \phi < 180°$[19]). While this is challenging but achievable in a 1D-OPA, it is practically unfeasible for a 2D-OPA (for which the reported smallest achieved pitch is $d = 3.6\lambda$[15]). Therefore, in order to scan the beam in θ as well as in $\phi$, a 1D-OPA must be used in conjunction with a second physical mechanism, unrelated to the phased array. Currently, by far the most widespread method relies on the wavelength dependence of the emission angle, $\theta$, of the diffraction grating used to extract the light from each waveguide[7–11,20]. Typically $d\theta/d\lambda = 0.05–0.1°/nm$, meaning that a 100 nm laser wavelength range can yield a scanning range in θ of 5–10°. However, in combination with other LIDAR-compatible laser requirements such as high output power (typically >100 mW), short pulse width (<10 ns for time-of-flight systems[21,22]) or narrow linewidths (<100 kHz for heterodyne detection schemes[23,24]) and fundamental mode operation (for efficient optical coupling to the photonic circuit), obtaining a laser that is efficient over such a wide spectral range is very challenging at low unit cost.

The ability to perform 2D beam steering at a single wavelength with a one dimensional OPA would enable the use of simpler, and therefore cheaper, source laser technology[25,26]. The requirement for broadband operation of the photonic integrated circuit as well as the laser-circuit coupling would also be removed. Single-wavelength, two dimensional beam-steering has been demonstrated using a switched multi-OPA approach[27,28], via the integration of a liquid-crystal filled cavities[29], by active tuning of the emitters[30] and using a single emitter on a thermally

Univ. Grenoble Alpes, CEA LETI, F38000 Grenoble, France. ✉ e-mail: daivid.fowler@cea.fr

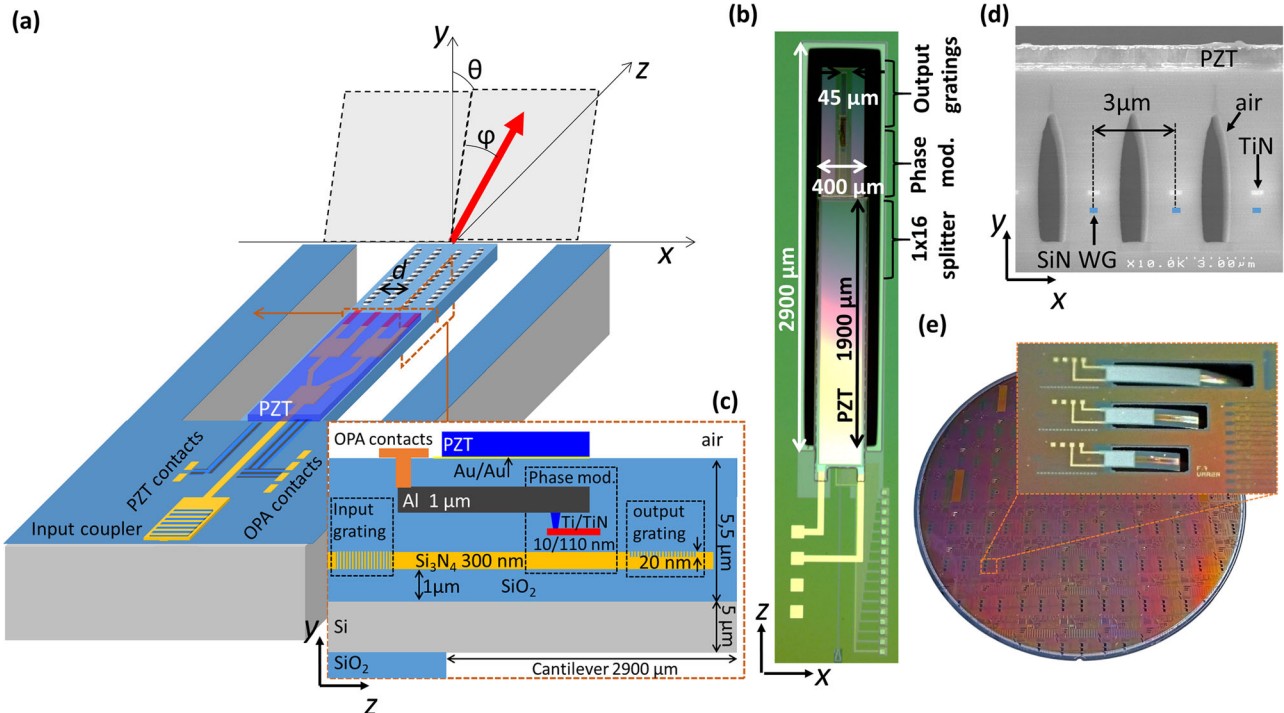

**Fig. 1 | Physical device characteristics. a** Schematic 3D view of the OPA on a MEMS cantilever. **b** Top-view image of the fabricated device. **c** Simplified schematic cross-section of the layer stack. **d** Scanning electron microscope image of TPM cross section showing adjacent waveguides (WGs) separated by air-filled voids. **e** 200 mm wafer containing fabricated devices.

**Table 1 | Comparison table of a selection of 2D beam steering demonstrations**

|  | φ/θ range (°) | φ/θ divergence (°) | λ (nm) | φ/θ mechanism |
|---|---|---|---|---|
| Poulton et al.[11] | 100/17 | 0.01/0.04 | 1490–1610 | OPA/Δλ |
| Inada et al.[29] | 15/16 | 7.6/0.16 | 940 | OPA/LCD |
| Fatemi et al.[15] | 16/16 | 0.8/0.8 | 1550 | OPA/OPA |
| Takeuchi et al.[30] | −/21 | −/0.2 | 1580 | -/thermal |
| Azadeh et al.[31] | 24/12 | 1.4/3.1 | 600 | MEMS/MEMS |
| This work | 17/40 | 1.2/0.65 | 905 | OPA/MEMS |

activated 2D MEMS cantilever[31]. Table 1 shows a comparison of a selection of demonstrations of solid-state 2D beam scanning devices using different physical mechanisms, highlighting the scanning range and beam divergence.

Currently, none of these demonstrated solutions offer single wavelength 2D beam scanning with a satisfactory combination of high frequency operation, wide angular range and low power consumption for demanding sensor applications. In this work, we demonstrate an OPA for which all the passive and active components, beyond the optical and electrical inputs, are embedded in a PZT (lead zirconate titanate) piezoelectric-actuated cantilever, as depicted schematically in Fig. 1a and as seen in the microscope top view in Fig. 1b. Using this device, fabricated entirely at the 200 mm wafer scale, we demonstrate continuous scanning in φ over 17° from a 1D-OPA and simultaneous beam scanning in θ over an angle, Δθ = 10° with a maximum applied PZT DC voltage of 20 V, and Δθ up to 40° at the beam resonant frequency of 1.52 kHz.

Although 2D-beam steering may be achieved using MEMS alone[31,32], MEMS elements with single-axis movements are less complex to fabricate, less sensitive to external mechanical perturbations and can achieve wider ranges of angular movement. Likewise, the enticing aspects of the integrated

OPA, as previously explained, are only achievable in a 1D format. The combination of 1D-MEMS and 1D-OPA therefore maximizes the advantages of each scanning mechanism.

Our work builds on many recent advancements that have shown the potential to expand the capabilities of integrated photonics via the integration of MEMS technologies[33]. We believe that the demonstrated combination of integrated active photonics and piezoelectric-actuated silicon MEMS technology, allowing single wavelength two-dimensional beam steering, provides a compelling pathway to a low-cost, OPA-based solid-state LIDAR system.

## Results

### Circuit architecture and operating principle

Figure 1a shows a schematic of the general features of the fabricated device. The 16 channel 1D-OPA (only 4 channels are shown for clarity) is designed to operate at λ = 905 nm to be compatible with commercial silicon-based photodetector technology[34]. The pitch of the output gratings, *d*, is 3 µm. The OPA circuit is based on single mode silicon nitride waveguides (propagation loss ~1 dB/cm) and consists of an input grating coupler (insertion loss (IL) ~ 5 dB) waveguide splitter tree, consisting of 4 rows of 1 × 2 power splitters (IL ~ 0.1 dB per splitter), which equally divides the input optical power into 16 waveguides. The phase of the light in each waveguide is then modulated individually by an array of thermal phase modulators (TPM). Each optical channel is then terminated by a shallow (20 nm) etched grating that diffracts the light of each waveguide into free space. The total fiber to free-space insertion loss is estimated to be around 8 dB. Beyond the optical and electrical inputs, which are fabricated on the fixed part of the photonic chip, all the above elements of the OPA circuit are embedded in the flexible cantilever, approximately of thickness, $t_{CL}$ = 11 µm, width, $w_{CL}$ = 400 µm and length, $l_{CL}$ = 2900 µm, which can be seen in the top view image in Fig. 1b.

Figure 1c shows a schematic cross-section of the principal elements of the layer structure (more details in the Methods sections), consisting of a 5 µm silicon support for mechanical rigidity, upon which the photonic

**Fig. 2 | Output beam properties and OPA beam steering. a** Camera image of the emission of an OPA on a cantilever calibrated at $\phi = 0°$ and for $V_{PZT} = 0$ V. **b** Cross-section in $\phi$ of the main beam profile. **c** Cross-sections in $\phi$ of OPA emission calibrated for various $\phi$ angles. **d** Cross-section in $\theta$ of the main beam profile. **e** Cross-section in $\theta$ of the main beam profile for an identical OPA not on a cantilever (fixed chip surface).

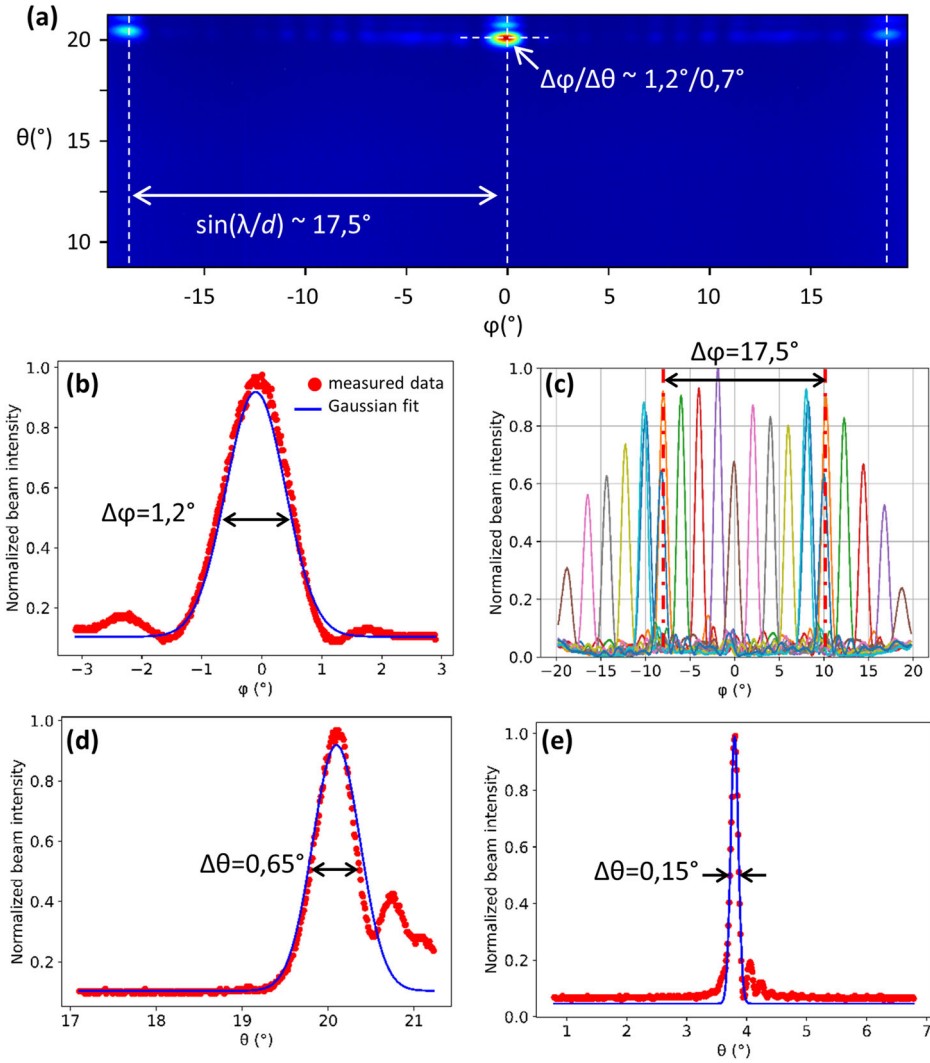

circuit is constructed, based on a patterned 300 nm-thick $Si_3N_4$ waveguide layer. TPMs, consisting of resistive titanium-nitride (TiN) wires placed above the waveguides, allow the phase of the light in each waveguide to be individually modulated. The TPMs are electrically connected to probe contacts on the fixed part of the chip via tungsten vias and a single layer of aluminum tracks. Following the completion of the photonic wafer fabrication, a PZT layer, prepared separately (so as not to exceed the thermal budget of the photonics technology) using a sol-gel technique on a 200 mm silicon wafer, is transferred to the photonic circuit using Au/Au wafer-to-wafer bonding[35,36]. The donor substrate is removed and the PZT layer is patterned[36,37] before the creation of electrical tracks and surface contact pads in order to apply a vertical electric field (in $y$). Finally the cantilever was created by selectively etching the $SiO_2$ around and beneath the desired cantilever shape (more details in the methods section). Figure 1d shows a scanning electron microscope cross-section image in the TPM section of the cantilever. This image shows key elements of the photonic circuit: the SiN waveguides (highlighted in blue), the TiN wires and air-filled voids providing thermal isolation between adjacent waveguides, and the superposed PZT layer. Figure 1e shows a complete 200 mm wafer containing the fabricated devices. Cantilevers of various lengths can be seen in the zoomed image.

## 2D beam scanning with quasi-static cantilever operation

The characterization of the OPA was performed at the wafer-level as described in previous work[38]. This avoids chip dicing as well as complex packaging, which greatly facilitates the evaluation of multiple circuits. Pulsed light from an external laser (~2 μs duration, ~25 Hz repetition rate) was coupled to the OPA via an optical fiber. The electrical signals applied to each of the 16 TPMs in the OPA were also pulsed (synchronously with the laser pulses) to avoid unnecessary heating of the cantilever, which would have led to thermally induced flexion of the cantilever and ultimately to irreversible thermal damage (mitigating measures are suggested in the discussion section below). The light emitted at the OPA output was visualized directly on the focal plane array of an imaging camera. Details of the measurement setup and characterization methods can be found in Supplementary Note 1.

With no voltage applied to the PZT actuator, the functionality of the photonic circuit was confirmed. Figure 2a presents the obtained image when configuring the OPA so that the 0th order main beam $\phi_0$ is centered at $\phi = 0°$. The divergence in $\phi$ of the main beam was measured to be $\Delta\phi = 1.2°$ (extracted from the Gaussian fit (blue line) in Fig. 2b), which is close to the theoretical value $\Delta\phi \sim \lambda/N_{ch}d \sim 1.25°$. By applying a set of voltages to the OPA contacts, the main beam can be centered at a chosen $\phi$ value. In Fig. 2c, the OPA emission profile is shown for a main beam calibrated in 2° steps in a window of $-10° < \phi_0 < 10°$. Each line color corresponds to a different $\phi_0$ value. The unambiguous scanning range, over which only the zeroth order main beam emits, is consistent with the theoretical value $\Delta\varphi = \sin^{-1}\lambda/d \sim 17.6°$.

The beam profile in $\theta$ is shown in Fig. 2d. The emission peak is found at $\theta = 20°$ relative to the wafer surface normal. The divergence of the beam is

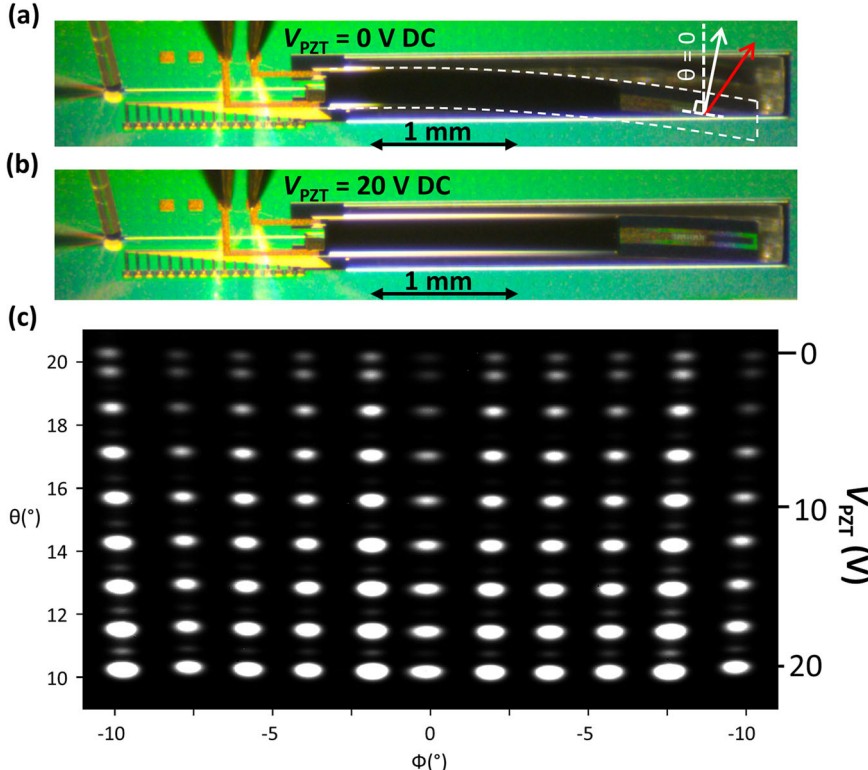

**Fig. 3 | 2D beam steering using the quasi-static cantilever regime. a** Microscope image of fabricated device with $V_{PZT} = 0$ V. **b** Fabricated device with $V_{PZT} = 20$ V. **c** Composite image of device optical output configured for various main beam emission in $\theta$ and $\phi$.

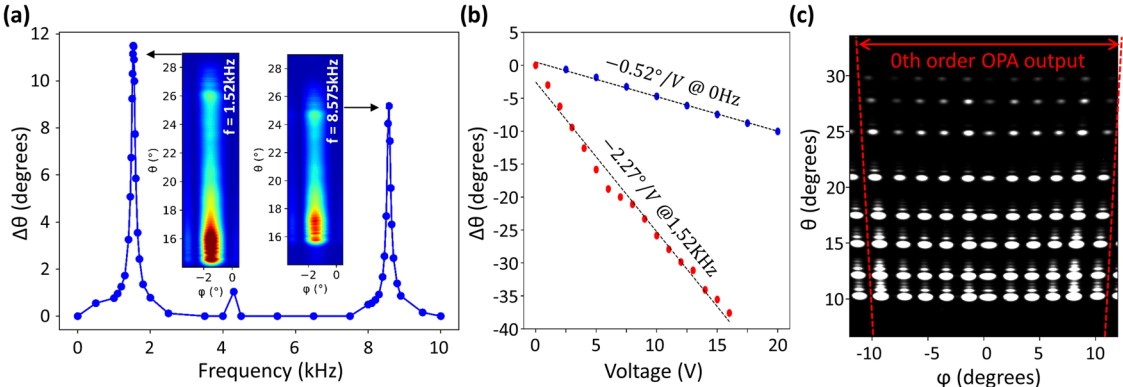

**Fig. 4 | 2D beam steering using the resonant cantilever regime. a** Amplitude of the beam displacement in $\theta$ as a function of the excitation signal frequency for $V_{max} = 4$ V. **b** Extracted beam scanning amplitude in the quasi-static and the resonant regime (at the fundamental resonant mode frequency $f_0$) as a function of the applied voltage $V_{max}$. **c** Composite image of the device optical output configured for various main beam emissions in $\theta$ and $\phi$ for resonant cantilever operation ($V_{max} = 6$ V).

$\Delta\theta = 0.65°$. Comparing with the $\theta$-beam profile (Fig. 2e) of an identical OPA placed on a fixed wafer surface (not on a cantilever) shows a clear difference in both the emission angle, $\theta = 4°$, and the divergence, $\Delta\theta = 0.15°$, which both approximately correspond to the expected antenna behavior. The close-up image of the fabricated device in Fig. 3a indicates the origin of the modified $\theta$-beam profile. It can be clearly seen that the cantilever tip is bent downwards towards the substrate, increasing the angle of the wafer surface normal vector (solid white arrow), thus accounting for the 16° difference in the emission angle in the $\theta$-direction. This deflection is due to compressive stress inherent in the multi-layered structure. We attribute the increase in the beam divergence to the curvature of the cantilever surface over the length of the output diffraction grating.

Applying a constant voltage, $V_{PZT} = 20$ V, visibly straightens the cantilever (Fig. 3b) due to the tensile force induced by the PZT layer (a moving image can also be seen in supplementary movie 1). The $\theta$-value of the wafer surface normal vector is reduced and the emitted beam $\theta$ angle declines at a

rate of 0.52°/V (Fig. 4b). In this quasi-static regime we may then scan the beam in the range $10° \leq \theta \leq 20°$. It should be noted that the absolute $\theta$ value of the cantilever is subject to hysteresis in the PZT actuator (more information in supplementary note 2). The movement of the cantilever in the quasi-static regime can be seen in supplementary movie 1.

Applying static voltages beyond 20 V can lead to dielectric breakdown of the PZT layer, which renders the device inoperable. The breakdown voltage is limited by the defect density in the PZT material and can be further increased by optimizing the PZT material quality and the film transfer process. Figure 3c shows a superposition of the beam images taken while simultaneously varying $\theta$ via $V_{PZT}$ and $\phi$ via the OPA operation. The calibration of the OPA for a given $\phi$ value remained valid for all $\theta$ values, indicating that the cantilever surface remained optically flat in the lateral direction of the emission surface during cantilever flexion, and that the induced strain does not perturb the photonic circuit components. The weak secondary spots visible vertically above the bright primary spots are caused

**Fig. 5 | Finite element method simulations of PZT actuated cantilevers.** Fundamental resonance mode at the maximally depressed state for (**a**) Cantilever width, $W_{CL} = W_{PZT} = 400$ μm and (**b**) $W_{CL} = W_{PZT} = 1000$ μm.

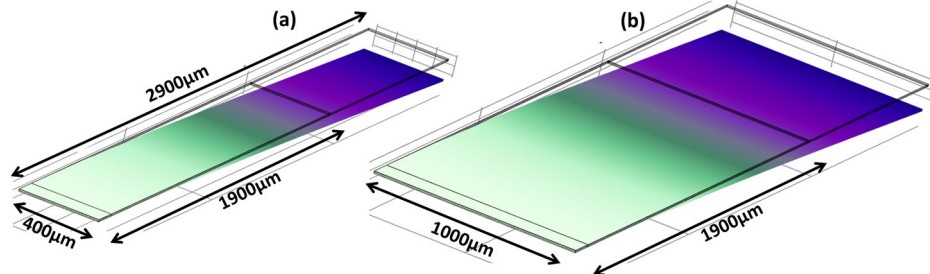

by the secondary emission peak in θ that can be seen in Fig. 2d. This can be avoided with a chirped antenna grating design[39]. In the quasi-static regime, the power consumption of the PZT actuator is ~1000× smaller than that of the TPMs, with leakage currents less than 1 μA (indicated on the PZT power supply at $V_{PZT} = 20$ V during quasi-static device operation). The reduction in beam amplitude for higher θ angles is due to a design error whereby the output beam is partially obscured by the unetched region opposite the distal cantilever tip (see Fig. 3a). This is further described in supplementary note 3.

## 2D beam scanning with resonant cantilever operation

While in the quasi-static regime the range of accessible beam θ values was limited by the dielectric breakdown of the PZT layer, much greater θ values may be obtained in a dynamic regime by applying a sinusoidal voltage $V_{PZT}(t) = V_{max}(1+\sin(2\pi f t))/2$ such that the cantilever enters a resonant mode. Figure 4a shows the amplitude of the variation of emission angle, Δθ, as a function of the excitation frequency, $f$, for $V_{max} = 4$ V. The fundamental resonant mode can be clearly observed at a frequency, $f_0 \sim 1.52$ kHz as well as a higher order resonant mode $f_1 \sim 8.5$ kHz. The color plot inserts corresponding to each resonance peak show long-exposure beam images, showing a smeared spot along the θ-direction, allowing the angular oscillation amplitude to be derived (see supplementary note 3).

With the excitation frequency fixed at $f_0$, it can be seen in Fig. 4b how Δθ varies as a function of $V_{max}$. For $V_{max} = 6$ V, Δθ ~ 20°, which is much greater than that observed in the quasi-static mode ($f = 0$ Hz). The maximum value of Δθ observed at $f_0$ was almost 42° at $V_{max} = 16$ V. As in the quasi-static mode, this was limited by the electrical breakdown of the PZT layer and not by mechanical degradation of the cantilever or the photonic or electronic circuitry contained within. By synchronizing the sinusoidal voltage applied to the PZT layer and the voltages applied to the OPA contacts, we performed raster scanning of the OPA beam. Figure 4c shows a composite image of camera images obtained using a stroboscopic capturing method (more details in supplementary note 1). The weak beam intensity observed for low θ values was again due to the beam being partially obscured by the fixed edge opposite the cantilever tip. The effect is exaggerated in the dynamic regime due to the increased cantilever movement.

## Discussion

By integrating an active photonic circuit on a piezoelectric actuated MEMS cantilever, we have demonstrated how 2D beam steering may be achieved at a single wavelength using a 1D-OPA. However, as described in the introduction, to exploit the potential offered by the integrated optical phased array to perform beam steering in a LIDAR system with simultaneously narrow beam divergence (<1 mRad), high main lobe power fraction (>0.8) and sweeping range (Δϕ > 90°), it is necessary to use an OPA with hundreds or thousands of emitters[6]. Doing so would imply the use of a cantilever with a greater width than that used in the current demonstration. To illustrate the scalability of the cantilever system, we have simulated, using COMSOL® finite element analysis software, the mechanical properties of the fabricated cantilever structure (with a simplified layer structure omitting the photonic circuit layers) with dimensions $L_{CL} = 2900$ μm, $L_{PZT} = 1900$ μm, $t_{CL} = 11$ μm, $W_{CL} = W_{PZT} = 400$ μm (as per the fabricated device), to compare with a similar cantilever with $W_{CL} = W_{PZT} = 1000$ μm. An OPA operating at λ = 905 nm whose emission covered the width of the latter

structure would have a main beam divergence of ~0.05°/1 mRad). Figure 5a shows the $W_{CL} = 400$ μm (as for the fabricated device) cantilever flexion in the maximally depressed (which corresponds to maximum beam θ value) state in the first resonance mode. The simulated fundamental resonant frequency for the $W_{CL} = 400$ μm is $f_0 = 1686$ Hz, which is in satisfactory agreement with the measured value, $f_0 = 1520$ Hz. Figure 5b shows a similar simulation for $W_{CL} = W_{PZT} = 1000$ μm in which a very similar $f_0$ value of 1698 Hz is obtained. This simulation result is consistent with the mechanical model in ref. 40 which shows that neither the resonant frequency, $f_0 \propto \frac{t_{CL}}{L_{CL}^2}(\frac{E}{\rho})^{1/2}$ (where ρ is the cantilever density and $E$ is the Young's modulus), nor the maximum tip displacement, $\delta_0 \propto \frac{d_{31}L^2}{t_{CL}^2} * V_0$ (where $d_{31}$ is the transverse piezoelectric coefficient and $V_0$ is the applied voltage), is dependent on the cantilever width provided $W_{CL} = W_{PZT}$. It is however likely that for larger $W_{CL}$ values, measures should be taken to ensure that the emission surface at the distal end remained optically flat (so as to avoid, for example, the increased beam divergence in θ observed in Fig. 2d). An opportunity for such optimization of the mechanical properties of the cantilever lies in the selection of the initial thickness of the underlying silicon layer (5 μm in this work), which could then be selectively patterned to create non-uniform rigidity along its length.

Furthermore, we can estimate how the power consumption of the PZT actuator might evolve as the cantilever width is increased. While in non-resonant operation, excluding the effect of leakage current caused by defects in the PZT dielectric, operation of the cantilever does not consume any electrical power, in resonant mode, the capacitive load should be considered. In this case, the power consumption can be estimated as $P_{PZT} = \varepsilon\varepsilon_0 \frac{L_{PZT}W_{PZT}}{t_{PZT}} V^2 f$. Electrical measurements show that the PZT permittivity, ε, varies between ~400 and ~1400 in the operating voltage range. In resonant mode operation for $L_{PZT} = 1900$ μm, $W_{PZT} = 1000$ μm, $t_{PZT} = 550$ nm, $V = 6$ V, $f = 1520$ Hz, $P_{PZT} \sim 1$ mW, which is probably minor from a system point of view. Beyond performance improvements from the use of a greater number of OPA channels, the beam steering performance could be improved with other, non-thermal, waveguide phase modulation mechanisms. At an operating wavelength of 905 nm, this could be achieved electro-optically via the heterogeneous integration of lithium niobate[41,42] or even by using the PZT to induce stress-optic phase modulation[43]. The use of such non-thermal phase modulators, which could be placed on or upstream of the cantilever, would also mitigate the problems associated with heat dissipation in the cantilever region, allowing continuous operation, if required. Furthermore, phase modulators capable of frequencies several orders of magnitude beyond the cantilever resonant frequency (e.g., MHz) would allow raster scanning using the resonant cantilever motion in the slow axis, giving access to image refresh rates easily beyond the 10–100 Hz range available in current products.

In conclusion, we have demonstrated continuous two-dimensional beam steering at a single wavelength by integrating an active 1D-OPA photonic circuit within a piezoelectric-actuated cantilever. This integration of existing silicon photonics and MEMS technologies, based entirely on established industrial-scale 200 mm silicon wafer processing, retains all the advantages of OPA-based uni-dimensional beam steering while avoiding the necessity of a widely tunable laser to sweep the beam across the second dimension. Single wavelength operation will lead to simplification of all the elements of an OPA-based LIDAR device, considerably reducing system

complexity and cost. More generally, the incorporation of complex photonic circuits onto piezoelectric-actuated moveable surfaces offers the possibility of novel device architectures for integrated sensors.

## Methods

### Device fabrication

The initial photonic fabrication process is similar to previous work[27] but performed on a silicon on insulator substrate with a silicon thickness of 5 μm (see the layer stack in Fig. 1c). This silicon layer is necessary to ensure sufficient rigidity of the cantilever after local substrate removal (more details below). The circuit is based on a patterned 300 nm thick $Si_3N_4$ layer, deposited using low-pressure chemical vapor deposition at 750 °C. The 600 nm-wide waveguides were patterned, creating fully-etched and shallow-etched (20 nm) diffraction gratings for the light input and output, respectively. The TPMs featured a 10/110 nm-thick, 500 nm-wide Ti/TiN heating element running vertically parallel to a straight 220 μm-long waveguide with 1000 nm of intervening $SiO_2$. The waveguide pitch in the TPM section was the same as that of the output gratings ($d = 3$ μm). This required that each heating element and associated waveguide be thermally isolated so as to operate with minimal crosstalk. To achieve this, air-filled trenches were created between adjacent TPMs[6]. However, due to the subsequent PZT integration process, it was necessary to protect these trenches with a cap formed using low-temperature oxide deposition, creating air-filled voids, in a manner similar to that used to create high-k regions in microelectronic industry[44], that in our case maintain the thermal properties of the open air trenches while inhibiting contamination from subsequent process steps. This structure can be seen in the scanning electron microscope cross-section image in Fig. 1d.

### PZT-photonic integration

The PZT MEMS sol–gel actuation process requires a crystallization temperature of 700 °C, which is beyond the thermal budget of the photonic circuit components (notably the metal contact lines), precluding a monolithic integration scheme. In this work, a PZT film from a 200 mm Silicon donor substrate was transferred to a completed 200 mm silicon host substrate. The preparation of the PZT layer was performed as follows. A 500 nm thick SiO2 and a 100 nm-thick Pt bottom electrode are first deposited onto a bulk silicon substrate. The PZT piezoelectric film is then deposited using a sol–gel chemical solution deposition method. The 0.535 μm thick PZT film is made of 10 layers of a commercial PZT (52/48) solution provided by Mitsubishi Materials Corporation. Each layer of PZT is automatically spun, dried (at 130 °C) and calcinated (at 360 °C). A first crystallization step is performed on the first deposited layer to promote (100) orientation, and to ensure optimal piezoelectric properties. Further crystallization steps are then performed every three layers at 700 °C for 1 min under $O_2$ in a rapid thermal annealing (RTA) furnace. Then a 50 nm thick Pt electrode layer is sputter deposited onto the PZT film, followed by 50 nm of Au as a bonding layer. The completed photonic wafer is covered with a $SiO_2$ layer (PECVD) followed by an intermediate bilayer, W (20 nm)/WN(50 nm) and finally a 50 nm thick Au bond layer. Both donor and receiver Si substrates are then bonded together by Au-Au thermo-compression at 50 °C/30 min. The quality of the bonding is checked by surface acoustic microscopy before the donor substrate is mechanically removed by inserting a blade between the bonded wafers at the weakly adhered $Pt/SiO_2$ interface. Following the PZT transfer, the top electrode, the PZT thin film, the bottom electrode and the Au–Au bilayers were successively patterned using Reactive Ion Etching and ion beam etching via separate lithography steps. Further $SiO_2$ passivation Au deposition and patterning allowed the creation of the top contact pads.

### Numerical simulations

The passive photonic circuit components (waveguides, fiber grating couplers, MMI splitters and output gratings) were designed using RSOFT FEMSIM/FULLWAVE optical modeling software. The thermal phase modulator design was optimized using LUMERICAL DEVICE software.

The cantilever mechanical design was carried out using COSMOL multiphysics software.

## Data availability

The data that support the findings of this study are available from the corresponding author upon reasonable request.

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

## Acknowledgements

The authors would like to thank our colleagues Thierry Pellerin, Amélie Bellemin-Comte, Léopold Virot and Bertrand Szelag for their stimulating discussions related to this work.

## Author contributions

Sylvain Guerber: Characterization lead, measurement hardware and software development. Results extraction and presentation, co-author. Daivid Fowler: Photonic circuit and component design, project coordinator, lead author. Laurent Mollard: Cantilever simulation, design and photonic/MEMS process integration. Christel Dieppedale: MEMS process development and implementation. Gwenael Le Rhun: PZT process integration. Antoine Hamelin: MEMS/photonics characterization and electronics. Jonathan Faugier-Tovar: Photonic process development and implementation. Kim Abdoul-Carime: Photonic control electronics design and implementation.

## Competing interests

The authors declare no competing interests.
