## [Peer Review File · Communications Engineering]

Reviewers' comments:

Reviewer #1 (Remarks to the Author):

The manuscript submitted by Guerber et al. presents an innovative approach for biaxial beam steering by a linear Optical Phased Array and piezo-MEMS actuated cantilever waveguide scanning for the 2 orthogonal scanning directions. The manuscript is overall well written, and the work is complete including introduction of design, manufacturing, methods, and experimental characterization, and discussion of the results. It is a valuable contribution to the community and I recommend publication, provided the following items can be addressed:

- The reference to Silicon Photonics (abstract and manuscript) is misleading and should be revised. In silicon photonics, silicon is used as waveguiding material. Here, silicon nitride is the waveguiding material.
- The choice of wavelength is not substantiated. Why 905nm?
- The technique of transferring sol-gel PZT by wafer-bonding and handle removal is elaborate. In the main text it is not evident why this method is chosen, the motivation cited in the methods section. I recommend to state the motivation briefly in the main text.
- It is briefly mentioned, that pulsed light was chosen to avoid unnecessary heating of the cantilever. Is the heating significant / can it be estimated? What are the issues with heating? How can this be avoided?
- The quality of Figure 3a, b is quite low in the pdf file. It would be good to have high quality image for the final version.
- Figure 3 c) 'shadow images' (faint circles) are visible. Please include a discussion.
- Hysteresis and drift of PZT actuator are not discussed.
- The statement of power consumption on p.6 is 'virtually inexistent' is misleading, in view of the reported leak current. The authors should revise the statement and provide an analysis of the power consumption. In order to compare the demonstrated achievement, it would be of benefit to benchmark with other demonstration combining integrated photonics and MEMS.
- The authors hypothesize that fast phase modulators would allow fast raster scanning. It is not discussed what modulation mechanism would be suitable and how it would be integrated.
- The MEMS based scanning in resonance is limited to 1.52 kHz. This appears low for raster-scanning in LIDAR. Please include a discussion, comparing to other 2D OPAs (thermal, MEMS, ...) and MEMS micro-mirrors.

Reviewer #2 (Remarks to the Author):

The manuscript by Guerber and Fowler et al. describes a novel architecture for an optical phased array (OPA) beam scanner that combines MEMS and phase shifts to achieve 2D beam scanning. The approach of combining silicon photonics with piezoelectric MEMS actuation is innovative. The manuscript presents an effective demonstration. I support the acceptance and publication of the manuscript after the authors address the following comments.

1. The introduction can be strengthened by discussing some challenges with MEMS for LiDAR and beam scanning. A concern is reliability and the sensitivity to mechanical disturbances.
2. At the bottom of page 2: 2D beam scanning with a 1D OPA can still lead to high order beams if the pitch is not sufficiently small. Please clarify.
3. Please include a clearer circuit schematic in Fig. 1. I could not see any 1x2 power splitter in the schematic, and the devices are too small in the die photo.
4. Details about the sol-gel parameters, annealing temperatures, thickness, yield, and reliability would offer valuable insights into the reproducibility of the device performance. Please provide the piezoelectric coefficient.
5. The description of the cantilever piezoelectric actuation should include more information on the voltage ranges used for the corresponding beam deflection angles (pages 4 to 5).
6. Include a comparison of the beam divergence and scanning range with existing technologies. A table will be clarifying.
7. How can the divergence angle of the beam be further reduced?
8. What resonant frequency would be needed for a LiDAR application? Are the achievable scan frequencies sufficiently high? If not, how can the scan frequencies be increased without sacrificing the beam aperture (i.e., divergence)?
9. Please include some basic specifications about the photonic circuits for the wavelength used (905 nm), such as the propagation loss, fiber-to-chip coupling loss, overall emission efficiency / emitted power, excess loss of the power splitter, and waveguide dimensions. Are these waveguides single mode?
10. In Fig. 2, are the blue lines in (b), (d) and (e) simulated results or fits? What are the expectations from simulations?
11. Similarly, for Fig. 4, what are the expectations from simulations?
12. The fabrication section mentions a 200mm silicon wafer scale. If possible, please elaborate on the yield, uniformity across the wafer, and how these factors were optimized.

Reviewers' comments:

Reviewer #1 (Remarks to the Author):

The manuscript submitted by Guerber et al. presents an innovative approach for biaxial beam steering by a linear Optical Phased Array and piezo-MEMS actuated cantilever waveguide scanning for the 2 orthogonal scanning directions. The manuscript is overall well written, and the work is complete including introduction of design, manufacturing, methods, and experimental characterization, and discussion of the results. It is a valuable contribution to the community and I recommend publication, provided the following items can be addressed:

- The reference to Silicon Photonics (abstract and manuscript) is misleading and should be revised. In silicon photonics, silicon is used as waveguiding material. Here, silicon nitride is the waveguiding material.

'silicon photonics' has been changed to 'silicon nitride photonics'.

- The choice of wavelength is not substantiated. Why 905nm?

"to be compatible with commercial silicon-based photodetector technology [34]" was added to explain this choice, together with a reference that discusses the use of the 905nm wavelength in LIDAR systems.

- The technique of transferring sol-gel PZT by wafer-bonding and handle removal is elaborate. In the main text it is not evident why this method is chosen, the motivation cited in the methods section. I recommend to state the motivation briefly in the main text.

"(so as not to exceed the thermal budget of the photonics technology)" was added in the main text.

- It is briefly mentioned, that pulsed light was chosen to avoid unnecessary heating of the cantilever. Is the heating significant / can it be estimated? What are the issues with heating? How can this be avoided?

"...which would have led to thermally induced flexion of the cantilever and ultimately to irreversible thermal damage (mitigating measures are suggested in the discussion section below)."

...

"the beam steering performance could be improved with other, non-thermal, waveguide phase modulation devices, which could be also placed on or upstream of the cantilever, which would also mitigate the problems associated with heat dissipation in the cantilever region"

- The quality of Figure 3a, b is quite low in the pdf file. It would be good to have high quality image for the final version.

We will provide the images with the highest possible resolution.

- Figure 3 c) 'shadow images' (faint circles) are visible. Please include a discussion.

The following text was added:

“The weak secondary spots visible vertically above the bright primary spots are caused by the secondary emission peak in ϑ that can be seen in figure 2d. This can be avoided with a chirped antenna grating design [1] “

- Hysteresis and drift of PZT actuator are not discussed.

Added in the main text

It should be noted that absolute ϑ -value of the cantilever is subject to hysteresis in the PZT actuator (more information in supplementary note 2).

More information regarding the electro-mechanical properties is added in a new supplementary note (2). Unfortunately, we do not have any data regarding the drift of the PZT actuator. We accept that this issue would have to be addressed in the system development stage as it has been in the numerous PZT-MEMS products on the market e.g. inkjet printer heads.

- The statement of power consumption on p.6 is ‘virtually inexistent’ is misleading, in view of the reported leak current. The authors should revise the statement and provide an analysis of the power consumption.

‘virtually inexistent’ has been changed to “~1000x smaller than that of the TPMs”

In order to compare the demonstrated achievement, it would be of benefit to benchmark with other demonstration combining integrated photonics and MEMS.

The following text is added with a reference to an excellent recent review article (N. Quack et al., “Integrated silicon photonic MEMS,” *Microsyst Nanoeng* 2023).

“Our work builds on many recent advancements that have shown the potential to expand the capabilities of integrated photonics via the integration of MEMS technologies [32]. »

We consider that a ‘benchmark’ is difficult to provide as our integration concept of placing the active circuit directly on the moveable membrane differs too greatly from the existing photonics/MEMS demonstrations.

- The authors hypothesize that fast phase modulators would allow fast raster scanning. It is not discussed what modulation mechanism would be suitable and how it would be integrated.

The discussion section has been modified to address this point

“Beyond performance improvements from the use of a greater number of OPA channels, the beam steering performance could be improved with other, non-thermal, waveguide phase modulation mechanisms. At an operating wavelength of 905nm, this could be achieved electro-optically via the heterogeneous integration of lithium niobate [2], [3] or even by using the PZT to induce stress-optic phase modulation [4]. The use of such phase modulators, which could be placed on or upstream of the cantilever, would also mitigate the problems associated with heat dissipation in the cantilever region. Furthermore, phase modulators capable of frequencies far beyond the cantilever resonant frequency

(e.g. MHz) would allow raster scanning using the resonant cantilever motion in the slow axis, giving access to high image refresh rates.”

- The MEMS based scanning in resonance is limited to 1.52 kHz. This appears low for raster-scanning in LIDAR. Please include a discussion, comparing to other 2D OPAs (thermal, MEMS, ...) and MEMS micro-mirrors.

We consider that if the phase modulation frequency can be made much faster than the beam resonant frequency, the OPA will perform the fast axis scan and the beam will perform in the slow axis. In this case, the beam resonant frequency must be greater than the desired refresh rate, which is generally 10-100Hz. The current resonant frequency is therefore largely sufficient. This is reflected in the following text:

Furthermore, phase modulators capable of frequencies several orders of magnitude beyond the cantilever resonant frequency (e.g. MHz) would allow raster scanning using the resonant cantilever motion in the slow axis, giving access to image refresh rates easily beyond the 10-100 Hz range available in current products.

Reviewer #2 (Remarks to the Author):

The manuscript by Guerber and Fowler et al. describes a novel architecture for an optical phased array (OPA) beam scanner that combines MEMS and phase shifts to achieve 2D beam scanning. The approach of combining silicon photonics with piezoelectric MEMS actuation is innovative. The manuscript presents an effective demonstration. I support the acceptance and publication of the manuscript after the authors address the following comments.

1. The introduction can be strengthened by discussing some challenges with MEMS for LiDAR and beam scanning. A concern is reliability and the sensitivity to mechanical disturbances.

The following text has been added.

Although 2D-beam steering may be achieved using MEMS alone [31,32], MEMS elements with single axis movements are less complex to fabricate, less sensitive to external mechanical perturbations and can achieve wider ranges of angular movement. Likewise, the enticing aspects of the integrated OPA, as previously explained, are only achievable in a 1D format. The combination of 1D-MEMS and 1D-OPA therefore maximizes the advantages of each scanning mechanism.

2. At the bottom of page 2: 2D beam scanning with a 1D OPA can still lead to high order beams if the pitch is not sufficiently small. Please clarify.

The following paragraph is changed to emphasize the fact that the pitch, $d < \lambda$ is necessary for a 1D-OPA.

However, unlike LED arrays and CMOS detector arrays that leverage dense opto-electronic integration to create large 2D arrays of emission/reception points, a phased array requires that the pitch, d , between the emission apertures of adjacent optical channels be less than the operating wavelength, λ (and as little as $d = \lambda/2$ to ensure a high power fraction emitted in the fundamental emission lobe for any main beam angle, $0 < \phi < 180^\circ$ [5]). While this is challenging but achievable in a 1D-OPA, it is practically unfeasible for a 2D-OPA. (the reported smallest achieved pitch is $d = 3.6\lambda$ [6]).

3. Please include a clearer circuit schematic in Fig. 1. I could not see any 1x2 power splitter in the schematic, and the devices are too small in the die photo.

Figure 1a has been modified to depict the 1x2 splitters underneath a transparent PZT layer.

4. Details about the sol-gel parameters, annealing temperatures, thickness, yield, and reliability would offer valuable insights into the reproducibility of the device performance. Please provide the piezoelectric coefficient.

The sol-gel parameters are in the methods section. A graph of the piezoelectric coefficient $d_{31}(V)$ has been added in supplementary note 2.

While the fabrication of this proof of concept device was carried out using industrial scale methods, unfortunately we do not yet have any data regarding the yield or reliability of the PZT

transfer process. We accept that the reliability and yield would have to be studied during industrial transfer, but that these subjects are beyond the scope of this proof-of-concept demonstration.

5. The description of the cantilever piezoelectric actuation should include more information on the voltage ranges used for the corresponding beam deflection angles (pages 4 to 5).

Figure 4b shows the correspondence between the deflection angle and the voltage in the quasi-static and resonant operation.

6. Include a comparison of the beam divergence and scanning range with existing technologies. A table will be clarifying.

A comparison table has been included in the introduction.

7. How can the divergence angle of the beam be further reduced?

The need for a high number of OPA channels is now reiterated more clearly in the discussion section.

However, as described in the introduction, to exploit the potential offered by the integrated optical phased array to perform beam steering in a LIDAR system with simultaneously narrow beam divergence (<1 mRad), high main lobe power fraction (>0.8) and sweeping range ($\Delta\phi > 90^\circ$),...

As for the divergence in the second dimension:

...measures should be taken to ensure that the emission surface at the distal end remained optically flat (so as to avoid, for example, the increased beam divergence in ϑ observed in figure 2d).

Furthermore, the cited reference [39] describes design strategies to reduce divergence in this dimension.

8. What resonant frequency would be needed for a LiDAR application? Are the achievable scan frequencies sufficiently high? If not, how can the scan frequencies be increased without sacrificing the beam aperture (i.e., divergence)?

The following text has been added to the discussion section.

Furthermore, phase modulators capable of frequencies several orders of magnitude beyond the cantilever resonant frequency (e.g. MHz) would allow raster scanning using the resonant cantilever motion in the slow axis, giving access to image refresh rates easily beyond the 10-100 Hz range available in current products.

We consider that if the phase modulation frequency is much faster than the beam resonant frequency, the OPA will perform the fast axis scan and the beam will perform in the slow axis. In this case the beam resonant frequency must be greater than the desired refresh rate, which is generally 10-100Hz. The current resonant frequency is therefore largely sufficient.

9. Please include some basic specifications about the photonic circuits for the wavelength used (905

nm), such as the propagation loss, fiber-to-chip coupling loss, overall emission efficiency / emitted power, excess loss of the power splitter, and waveguide dimensions. Are these waveguides single mode?

Information has been added to the following paragraph:

The OPA circuit is based on single mode silicon nitride waveguides (propagation loss $\sim 1\text{dB/cm}$) and consists of an input grating coupler (insertion loss (IL) $\sim 5\text{dB}$) waveguide splitter tree, consisting of 4 rows of 1×2 power splitters (IL $\sim 0.1\text{dB}$ per splitter), which equally divides the input optical power into 16 waveguides. The phase of the light in each waveguide is then modulated individually by an array of thermal phase modulators (TPM). Each optical channel is then terminated by a shallow (10nm) etched grating that diffracts the light of each waveguide into free space. The total fibre to free-space insertion loss is estimated to be around 8 dB

The waveguide dimensions are given in the methods section. More information is available in ref [27].

10. In Fig. 2, are the blue lines in (b), (d) and (e) simulated results or fits? What are the expectations from simulations?

The blue lines are Gaussian fits of the measured data. A legend has been added in figure 2b to indicate this.

The text now comments more explicitly on the comparison with expected values.

... $\Delta\phi = 1.2^\circ$ (extracted from the Gaussian fit (blue line) in figure 2b), which is close to the theoretical value $\Delta\phi \sim \lambda/N_{\text{ch}}d \sim 1.25^\circ$...

...Comparing with the ϑ -beam profile (figure 2e) of an identical OPA placed on a fixed wafer surface (not on a cantilever) shows a significant difference in both the emission angle, $\vartheta = 4^\circ$, and the divergence, $\Delta\vartheta = 0.15^\circ$, which both approximately correspond to the simulated antenna behavior.

11. Similarly, for Fig. 4, what are the expectations from simulations?

The following comparison is present in the discussion section

"The simulated fundamental resonant frequency for the $W_{\text{CL}} = 400 \mu\text{m}$ is $f_0 = 1686 \text{ Hz}$, which is in satisfactory agreement with the measured value, $f_0 = 1520 \text{ Hz}$ "

12. The fabrication section mentions a 200mm silicon wafer scale. If possible, please elaborate on the yield, uniformity across the wafer, and how these factors were optimized.

While the fabrication of this proof of concept device was carried out using industrial scale methods, unfortunately we do not yet have any data regarding the yield or reliability of the PZT transfer process. Please also see the response to comment 4.

Reviewers' comments:

Reviewer #1 (Remarks to the Author):

The authors have addressed all queries and I recommend the manuscript to be published in the current form.

In response to the query on power consumption, the authors have provided a short comparison of PZT actuation to TPMs: "In the quasi-static regime, the power consumption of the PZT actuator is ~1000x smaller than that of the TPMs, with leakage currents less than 1 μA ". It is not evident, if the power consumption was measured experimentally and on which device. As the low power consumption appears to be a clear benefit of the demonstrated method, it would further strengthen the manuscript, if the authors could elaborate in an optional minor revision more in detail on the power consumption. What is the experimentally observed power consumption, and how was it measured?

Reviewer #2 (Remarks to the Author):

The authors have addressed the comments in the last review.

Reviewer #1 (Remarks to the Author):

The authors have addressed all queries and I recommend the manuscript to be published in the current form.

In response to the query on power consumption, the authors have provided a short comparison of PZT actuation to TPMs: "In the quasi-static regime, the power consumption of the PZT actuator is $\sim 1000\times$ smaller than that of the TPMs, with leakage currents less than $1\ \mu\text{A}$ ". It is not evident, if the power consumption was measured experimentally and on which device. As the low power consumption appears to be a clear benefit of the demonstrated method, it would further strengthen the manuscript, if the authors could elaborate in an optional minor revision more in detail on the power consumption. What is the experimentally observed power consumption, and how was it measured?

To clarify that this leakage current was measured on the operational device, the following information is added.

This can be avoided with a chirped antenna grating design [39]. In the quasi-static regime, the power consumption of the PZT actuator is $\sim 1000\times$ smaller than that of the TPMs, with leakage currents less than $1\ \mu\text{A}$ (indicated on the PZT power supply at $V_{\text{PZT}} = 20\ \text{V}$ during quasi-static device operation).

Reviewer #2 (Remarks to the Author):

The authors have addressed the comments in the last review.